# Research Progress on the Efficacy and Mechanism of Acupuncture in Treating Chronic Gastritis

**DOI:** 10.3390/diseases13110363

**Published:** 2025-11-07

**Authors:** Jing He, Hongye Wang, Cong Che, Anjie Wang, Ru Nie, Jinghong Tan, Jialin Jia, Zijian Liu, Tie Li, Guojuan Dong

**Affiliations:** Department of Acupuncture and Tuina, Changchun University of Chinese Medicine, Changchun 130117, China; hejing199801@163.com (J.H.); 17743196136@163.com (H.W.); 15005525978@163.com (C.C.); wanganjie201312@163.com (A.W.); nieru19981229@163.com (R.N.); 15973360536@163.com (J.T.); jiajialin0416@126.com (J.J.); 17604207415@163.com (Z.L.)

**Keywords:** chronic gastritis, atrophic gastritis, mechanism, acupuncture, electroacupuncture, moxibustion

## Abstract

**Simple Summary:**

Chronic gastritis (CG) is a common gastrointestinal disorder. Its symptoms often progress over time. Conventional treatments focus on anti-infective therapy and gastric mucosal protection. However, these approaches can cause adverse reactions with long-term use. external traditional Chinese medicine (TCM) therapies offer promising alternatives. These include acupuncture, electroacupuncture, and moxibustion. This review explains the mechanisms of acupuncture in treating CG. Acupuncture helps repair the gastric mucosa and restore gut microbiota balance. It also alleviates inflammation and regulates metabolic processes. Clinical studies show that specific acupoints are frequently used. Common examples include ST36, CV12, and ST21. Future research should investigate the complex regulatory network of acupuncture. Multidimensional analyses using qualitative, quantitative, and localist approaches are needed. Furthermore, standardizing clinical trials with multicenter, large-sample designs is crucial. This will provide robust scientific evidence for the efficacy of acupuncture against CG.

**Abstract:**

Chronic gastritis (CG) is a prevalent digestive disorder. It progresses through multiple stages, has an insidious onset, and can lead to severe complications if untreated. Modern treatments primarily aim to eradicate *Helicobacter pylori* and relieve symptoms. However, drug resistance and adverse effects often limit their effectiveness. As a primary traditional Chinese medicine (TCM) therapy, acupuncture treats CG through multi-target mechanisms. This review systematically outlines the classification and pathology of CG. It also comprehensively analyzes animal and clinical studies on acupuncture for CG from the past decade. The study summarizes the mechanisms of acupuncture and related therapies for CG, covering gastric mucosal function, metabolism, intestinal flora, gastrointestinal hormones, apoptosis, inflammation, and oxidative stress. It further explores the relationships among diseases, interventions, acupoints, and molecular pathways. Additionally, it compares the therapeutic profiles of different external therapies. The review also examines the current state of clinical research, including the selection of acupoints, treatment duration, and outcome assessment. The results demonstrate that external therapies effectively alleviate common CG symptoms such as abdominal distension, acid reflux, and stomach pain. These treatments also improve gastric mucosal health and modulate serum levels of inflammatory factors, oxidative stress markers, and gastrointestinal hormones. In vivo experiments using chronic non-atrophic gastritis (CNAG) and chronic atrophic gastritis (CAG) models confirm these benefits, showing changes in key biomarkers and elucidating potential mechanisms. Nevertheless, future high-quality, large-sample clinical trials are still needed to firmly establish efficacy. Further mechanistic studies are also needed to validate the interconnections among relevant signaling pathways.

## 1. Introduction

Chronic gastritis (CG) is fundamentally characterized by gastric mucosal inflammation or atrophy. The condition is often asymptomatic. Symptomatic presentations typically include persistent or recurrent upper abdominal pain, bloating, and epigastric fullness [1]. Multiple factors contribute to CG pathogenesis, such as genetics, diet, environment, and infection. *Helicobacter pylori* (*H. pylori*) infection represents a significant risk factor for both CG and gastric cancer [2]. Epidemiologically, nearly half of the global population is infected with *H. pylori* [3,4]. Regional prevalence varies, affecting about one-third of individuals in North America and substantially higher proportions in Europe, South America, and Asia [5]. In China, the reported infection rate ranges from 40.6% to 55.8% [6,7]. Furthermore, large-scale endoscopic studies report a CG prevalence of nearly 90% [8]. Current Western management strategies primarily target symptom relief. Standard pharmacotherapy involves antibiotics, proton pump inhibitors, bismuth preparations, and acid-suppressing drugs. Despite their efficacy in symptom control, limitations persist, including prolonged treatment duration, frequent recurrence after discontinuation, and gastrointestinal adverse effects associated with their use.

Traditional Chinese Medicine (TCM) treatments are tailored for individual patient conditions, enabling comprehensive physiological regulation. Acupuncture serves as a primary TCM modality, and clinical studies confirm its therapeutic benefits for CG [9,10]. As a form of physical stimulation, acupuncture may involve multi-pathway regulatory processes. Its mechanism is characterized by multi-target interactions, initiated from acupoint stimulation and progressing through effect activation to target regulation. However, clinical and mechanistic research on acupuncture for CG remains limited. Furthermore, a significant lack of large-scale, high-quality randomized controlled trials (RCTs) investigating its efficacy persists. We systematically searched seven Chinese and English databases, including PubMed, Web of Science, and China National Knowledge Infrastructure (CNKI), for publications released between 1 January 2015 and 31 August 2025. Our search strategy used a combination of keywords and Medical Subject Headings (MeSH), targeting key terms such as “chronic gastritis,” “chronic atrophic gastritis,” “chronic non-atrophic gastritis,” “acupuncture,” “electroacupuncture,” and “moxibustion.” We included studies in all languages to minimize language bias. A total of 1216 records were initially identified. To address duplicate reporting, we first applied automated deduplication, then manually compared study details. For publications derived from the same cohort, we selected the version with the most comprehensive data or the most detailed methodology. After a two-stage screening process based on abstracts and full texts, 162 studies were included, comprising 91 clinical trials and 71 animal experiments. The study selection process is summarized in Appendix A, and full search strategies are provided in Appendix A. This review seeks to offer novel insights into the mechanisms of acupuncture for CG and to strengthen the evidence base for its clinical application.

## 2. Classification and Pathogenesis of CG

### 2.1. Classification of CG

Based on endoscopic and pathological findings, CG is classified into non-atrophic (superficial) and atrophic gastritis. Its progression often follows a sequence from normal gastric mucosa to chronic non-atrophic gastritis, atrophic gastritis, intestinal metaplasia, dysplasia, and eventually gastric cancer. Chronic non-atrophic gastritis (CNAG) is a common digestive condition marked by non-atrophic inflammatory lesions of the gastric mucosa. Endoscopic features frequently include mucosal erythema, hemorrhagic spots or plaques, edema, congestion, and exudates. Epidemiological studies show that CNAG occurs across all age groups, with prevalence rising with age. It is identified in approximately 42–78% of individuals undergoing gastroscopy [11,12]. Chronic atrophic gastritis (CAG) develops progressively from damage to the gastric mucosal epithelium and thinning of the mucosal layer, resulting in the loss or reduction in intrinsic glands.

### 2.2. Pathogenesis of CG

CG is characterized by gastric mucosal inflammation with a complex pathogenesis. Key mechanisms include gastrointestinal hormone dysregulation, inflammatory responses, oxidative stress, disturbances in gut microbiota, cellular aging, and apoptosis. Inflammation represents a coordinated response to pathogens, toxins, or tissue injury. This process engages diverse immune cells, signaling pathways, and molecular mediators. Upon activation, immune cells amplify gastric inflammation through upregulated inflammatory factors [13]. Reactive oxygen species (ROS) activate macrophages and inflammatory transcription factors. This stimulation promotes pro-inflammatory cytokine release, including interleukin (IL)-6, IL-1β, and tumor necrosis factor (TNF)-α. Oxidative stress arises from excessive ROS accumulation, disrupting cellular signaling and worsening inflammatory damage [14]. Critically, oxidative stress and inflammation maintain a self-perpetuating cycle: ROS drive inflammatory responses, while inflammation generates additional ROS [15]. Oxidative stress significantly contributes to the progression of CG. During *H. pylori* infection, immune-generated ROS intended to eliminate pathogens often cause collateral tissue damage [16]. These reactive molecules also impair the gastric mucosa through enhanced lipid peroxidation and compromised antioxidant defenses. Furthermore, they activate transcription factors, including nuclear factor-kappa B (NF-κB) and signal transducer and activator of transcription 3 (STAT3), thereby sustaining chronic inflammation through prolonged production of pro-inflammatory cytokines [17]. Persistent inflammation disrupts gastric function and hormonal secretion, particularly gastrin. This leads to reduced acid production and compromised mucosal protection. gastrin (GAS) stimulates acid secretion, activates digestive enzymes, and provides protection against oxidative stress and apoptosis. Inflammatory mediators, combined with low gastric acidity, disrupt intestinal motility, promote bacterial overgrowth, and exacerbate mucosal pathology, ultimately leading to the characteristic gastrointestinal symptoms [18].

The gastrointestinal microbiota possesses substantial biosynthetic capacity, generating molecules that modulate gastrointestinal inflammation. Gastric microorganisms include Lactobacillus and *H. pylori*. Certain Lactobacillus strains may suppress gastric acid and GAS secretion through mucosal acidification, potentially accelerating mucosal atrophy, intestinal metaplasia, and tumor development [19]. *H. pylori* establishes persistent colonization by evading host immunity, causing mucosal damage. Bile reflux and intestinal microbiota migration alter gastric microbial composition, elevate pH levels, and increase the presence of lipopolysaccharide-producing bacteria, thereby promoting gastric inflammation [20]. Gut microbiota contributes to disease pathogenesis through metabolic regulation. Microbial metabolites, including short-chain fatty acids and amino acids, modulate energy metabolism, inflammation, and intestinal barrier integrity [21]. Compromised intestinal permeability facilitates translocation of harmful substances (e.g., bacteria, endotoxins) into circulation, triggering immune and inflammatory responses in the gastric mucosa [22]. CAG involves cellular loss and aberrant differentiation, leading to mucosal senescence. Cell death processes—apoptosis, autophagy, and ferroptosis—are integral to the pathogenesis of CG. *H. pylori* infection induces apoptosis and inflammasome activation through bacterial toxins. Although autophagy serves as a cellular protection mechanism, its dysregulation exacerbates mucosal injury. *H. pylori* virulence factors can either induce autophagic cell death or inhibit autophagy to enhance inflammation [23]. Ferroptosis, an iron-dependent form of cell death, also contributes to the progression of disease. Elevated hepcidin induces gastric ferroptosis, resulting in impaired iron absorption, lipid peroxidation, and abnormal acid secretion [24].

## 3. Mechanism of Action of Acupuncture in Treating CG

Acupuncture shows potential for preventing and treating digestive diseases. With growing international recognition of acupuncture therapy, researchers have conducted numerous mechanistic studies on its application for CG. This section examines current research on the effects of this section on gastric mucosal function, metabolism, intestinal microbiota, gastrointestinal hormones, cell apoptosis and proliferation, and anti-inflammatory and antioxidant responses (Figure 1).

### 3.1. Regulate Stomach and Gastric Mucosal Function

Gastric smooth muscle contraction and relaxation, along with gastric acid secretion, play crucial roles in food digestion. The stomach accomplishes digestion through mechanical and chemical processes, with gastric motility representing a key mechanical component [25]. Under physiological conditions, gastric peristalsis begins when food enters the stomach. The antrum performs grinding and propulsion functions, rhythmically moving food toward the pylorus and duodenum [26,27]. Therefore, antral contraction frequency and gastric emptying serve as important indicators of gastric motility. Furthermore, hormones such as GAS, motilin (MOT), and ghrelin are closely associated with gastric motility [28]. In gastric diseases, mucosal defensive functions may be impaired before motility changes occur, while digestive functions remain largely preserved. Consequently, assessing gastric mucosal function and pathology provides another crucial indicator for evaluating gastric health. Preclinical studies demonstrate that acupuncture may treat CAG and precancerous lesions of gastric cancer (PLGC) by regulating gastric and mucosal functions. These therapeutic effects include enhanced gastric motility, regulated gastric electrical rhythm, improved mucosal blood flow, and optimized secretory functions, ultimately promoting mucosal regeneration and repair (Table 1).

Studies demonstrate that gastric antral motility index (MI) and gastric emptying are significantly reduced in CAG model rats. Animal research has shown that a two-week moxibustion treatment significantly improves these parameters and enhances gastric motility [29]. Furthermore, moxibustion effectively improved gastrointestinal motility in CAG rats by regulating ghrelin and its receptor, the growth hormone secretagogue receptor (GHSR), within the brain–gut axis [30]. Ghrelin, a growth hormone secretagogue, is a brain–gut peptide that is distributed throughout the gastrointestinal tract and the central nervous system. It travels through the bloodstream to distant tissues, including the hypothalamus and hippocampus, where it binds to GHSR. This binding plays a vital role in regulating gastrointestinal motility through both peripheral and central pathways. Electrogastrogram parameters primarily include the percentage of normal slow waves, gastric electrical frequency, and the percentage of gastric dysrhythmia, which effectively reflect gastric muscle bioelectrical activity and are associated with various gastric physiological and pathological phenomena. Luo Wei et al. [31] demonstrated that electroacupuncture (EA) combined with an Intelligent Meridian-Unblocking Therapy Device effectively regulates abnormal gastric electrical rhythms in CAG rats. This intervention significantly increased the average waveform frequency and amplitude, while reducing both the dysrhythmia index and the frequency coefficient of variation.

Based on animal studies, the regulatory effects of acupuncture on the gastric mucosa can be summarized in three main aspects: (1) Promotion of Mucosal Regeneration and Repair: Laser acupuncture delivers energy through localized acupoint irradiation, significantly improving gastric mucosal conditions in CAG rats. Treated animals exhibit smoother mucosal surfaces with distinct folds and significantly increased mucosal thickness compared to model controls [32]. Animal studies associated this improvement with modulated Trefoil Factor (TFF) expression—specifically enhanced TFF1 and suppressed TFF3 [33]. TFF peptides play crucial roles in gastrointestinal protection and repair, while maintaining low expression in normal mucosa [34]. Heat-sensitive moxibustion at CV12 demonstrates similar efficacy, upregulating growth hormone (GH) and Pepsinogen I (PGI) to improve mucosal morphology through enhanced circulation and glandular regeneration [35]. Moxibustion’s reparative effects involve the modulation of inflammatory factors at ST36, where localized stimulation initiates systemic immune responses that activate endogenous repair mechanisms [36,37]. (2) Regulation of Gastric Mucosal Blood Flow: Impaired mucosal perfusion represents a key pathological feature of CAG. The application of “Yu Han Nuan Wei Gao” plaster to stomach meridian acupoints effectively restored blood flow in CAG model rats, concurrently reducing pathological scores and delaying glandular degeneration [38]. Vascular Endothelial Growth Factor (VEGF) and Hypoxia-Inducible Factor-1α (HIF-1α) form critical regulatory axes for angiogenesis, with their overexpression exacerbating mucosal damage. Systematic reviews confirmed that acupoint catgut embedding at BL20, ST36, and CV12 significantly downregulates HIF-1α/VEGF expression, alleviating mucosal congestion and counter-acting atrophy progression [39]. (3) Regulation of Gastric Mucosal Secretion: PGI and PGII serve as valuable indicators of mucosal secretory function, reflecting the status of fundic and pyloric glands, respectively [40]. Gastrin-17 (G-17) provides a reliable assessment of antral mucosa status through its correlation with gastric acid secretion [41]. The severity of mucosal atrophy inversely correlates with G-17, PGI, and PGI/PGII ratio (PGR). Laser irradiation at ST36 facilitated parietal cell recovery in CAG rats and enhanced acid secretion through structural restoration [42]. Combined acupuncture at “He-Mu” points upregulated PGI, PGR, and G-17 levels, while heat-sensitive moxibustion at CV12 elevated serum GH—the primary trophic hormone for mucosal maintenance [35]. Yang Gaiqin’s research demonstrated that Back-Shu acupuncture significantly outperformed conventional drugs in elevating serum PGI, PGII, and GAS, while reducing MOT levels [43,44].

Research has established connections between diseases, interventions, acupoints, and molecular mechanisms (Figure 2). Subsequent experimental studies on acupuncture’s regulation of gastric function have consistently focused on CAG pathways. Acupoint selection demonstrates remarkable consistency, with ST36 and CV12 emerging as the most frequently utilized points. The regulation of gastric motility, electrical rhythms, and mucosal function ultimately promotes mucosal repair, primarily through modulating molecular expression, including prostaglandins (PGE2, PGF2α), mucins (MUC1, MUC5AC, MUC6), trefoil factors (TFF1, TFF3), and growth factors such as VEGF and Epidermal Growth Factor (EGF). Different intervention methods employ distinct pathways: laser acupuncture promotes gastric mucosal repair through TFFs [33], while acupoint catgut-embedding achieves similar effects via the HIF-α/VEGF pathway [39].

**Table 1 diseases-13-00363-t001:** Acupuncture Regulates Gastric and Gastric Mucosal Function in the Treatment of CG.

Objective	Disease	Animals	Intervention		Acupoint	Course of Treatment	Molecular Mechanism		References
Improve gastric motility	CAG	SD rats	Moxibustion	Moxa stick 0.8 cm × 12 cm	CV12	40 min QD d1–28		↑: Ghrelin, GHSR	[30]
				Smokeless moxa stick 0.4 cm × 12 cm	ST36	30 min QD d1–6,7d cycle, repeat × 4		↑: Gastric antral motility index, gastric emptying rate, MUC1, MUC5AC, MUC6	[29]
Regulate the state of the gastric mucosa				Moxa stick 0.8 cm × 12 cm	CV12	40 min QD d1–28		↑: GH, PGⅠ, PGⅡ, PGR	[35]
Regulate gastric electrical rhythms			EA and TENS	Continuous wave, frequency 50 Hz, intensity 2–5 V	ST36,CV12,ST25,BL20	20 min QD × 2 months		↑: PGE2, PGF2a	[31]
Promote mucosal regeneration and repair		Wistar rats	LA	Output power: 30 mW Wavelength: 632.8 nm Spot diameter: 2 cm Spot center aligned with acupoint Fiber tip distance from skin: 10 cm	ST36	5 min QD d1–14	↓:TFF3	↑: TFF1	[33]
Promote mucosal regeneration and repair		SD rats	ACE	Size 6 disposable syringe needle,0000 chromium-plated catgut suture	ST36,CV12	Q10D × 6	↓:HIF-α,VEGF		[39]
		SD rats			BL17,BL18,BL20,BL21,BL23	15 min QD d1–30		↑: PGI, PGⅡ	[44]
Increase gastric mucosal blood flow	PLGC		AA		ST36,ST21	1 h QD		↑: Gastric mucosal blood flow	[38]

Disease Corresponding Abbreviations: Chronic Atrophic Gastritis (CAG), Precancerous Lesions of Gastric Cancer (PLGC).

### 3.2. Regulation of Gastrointestinal Hormones

Gastrointestinal hormones comprise biologically active polypeptides secreted by endocrine cells and neurons of the enteric nervous system. Based on chemical structure, they are categorized into five families: the Gastrin-cholecystokinin (GAS-CCK) family, the Secretin family (including Secretin, glucagon, glicentin, Vasoactive Intestinal Peptide [VIP], and Gastric Inhibitory Polypeptide [GIP]), the Somatostatin (SS) family, the MOT family (including MOT, ghrelin, and obestatin), and Substance P (SP). These hormones regulate the motility, secretion, and absorption of the digestive organs through endocrine, paracrine, and neurocrine mechanisms. They facilitate the movement of colonic content and maintain homeostasis of the brain–gut axis. Preclinical studies demonstrate that acupuncture promotes gastric mucosal repair and regeneration in both CAG and CNAG. This therapeutic effect involves modulating gastrointestinal hormones, with the regulated hormone profiles remaining broadly consistent across studies (Figure 3, Table 2).

(1)GAS: Acupuncture intervention primarily upregulates GAS expression. G cells in the gastric antrum synthesize and secrete this hormone. Gastrin’s key physiological functions include directly and indirectly stimulating the secretion of gastric acid and pepsin, as well as promoting the proliferation, differentiation, and angiogenesis of gastric mucosal epithelial cells. Additionally, gastrin enhances gastrointestinal motility and accelerates gastric emptying [45].(2)MOT: Acupuncture produces effects opposite to its action on GAS, downregulating both MOT levels. These peptides stimulate gastrointestinal motility and accelerate the process of emptying. Elevated MOT enhances gastric contractions, but sustained intense smooth muscle compression may compromise gastric wall vasculature, reducing mucosal blood flow and causing ischemic injury. Similarly, increased MOT contributes to pyloric dysfunction and gastroduodenal discoordination, delaying gastric emptying and prolonging mucosal exposure to inflammatory stimuli [46].(3)SS: This hormone is mainly secreted by D cells in the gastric antrum. SS preserves gastric mucosal protection by maintaining non-protein-bound sulfhydryl groups through glutathione reductase activity [47]. Additionally, it inhibits both gastric acid secretion and GAS release, establishing a coordinated axis between GAS, SS, and gastric acid that helps maintain normal gastrointestinal function. Studies demonstrate that in rats with CAG, glandular atrophy and a reduction in the gastric antrum lead to decreased levels of GAS and SS. However, heat-sensitive moxibustion can enhance SS expression to support mucosal repair [48].(4)CCK: This gastrointestinal hormone and neuropeptide plays a crucial role in regulating digestive and nervous system functions. CCK significantly inhibits both solid and liquid gastric emptying [49]. It acts through peripheral and central pathways to delay gastric emptying. Both physiological and pharmacological CCK concentrations inhibit postprandial gastric emptying, with higher CCK levels correlating with slower rates of emptying [50]. Experimental evidence suggests that acupoint application therapy downregulates CCK expression, modulates gastrointestinal motility, and enhances gastric mucosal protection, thereby facilitating the repair of injury [51].(5)PGI, PGR, and G-17: Studies demonstrate that reduced PGI, PGR, and G-17 levels serve as biomarkers for gastric corpus and antrum atrophy. G-17, secreted by G cells, enhances gastric mucosal blood flow. PG, comprising PGI and PGII, reflects the quantity of gastric glands and pepsin secretion, indicating the mucosal status and function. Studies confirm that reduced PGR elevates gastric cancer risk, even in patients without mucosal atrophy [52]. Research shows that acupuncture significantly improves the general condition and mucosal tissue in rats with CAG by regulating PGI, PGR, and G-17 levels. Furthermore, multi-acupoint combinations prove more effective than single-point therapy for CAG [53].(6)Others: SP, a neuropeptide widely distributed in the enteric nervous system and gastrointestinal tract, serves as the primary excitatory neurotransmitter regulating gastrointestinal motility. It strongly stimulates gastrointestinal smooth muscle contraction, accelerating motility and gastric emptying [54]. Ghrelin, initially identified from human and rat stomachs, represents a brain–gut peptide primarily produced in the stomach [55]. This hormone promotes food absorption and gastric emptying while regulating energy expenditure and providing gastrointestinal protection and healing. Research demonstrates that acupoint application significantly upregulates both SP and Ghrelin levels in the gastric mucosa of rats with CAG [51]. Meanwhile, EGF counteracts pepsin-induced damage, suppresses excessive acid secretion, and exhibits anti-inflammatory and analgesic properties, thereby protecting the gastric mucosa. Mechanistic studies suggest that acupressure at Back-shu points may function through regulating EGF expression [43].

In summary, the gastrointestinal hormone network represents a crucial therapeutic target for acupuncture in treating CAG. The core acupoint combinations—including ST36, CV12, BL20, and BL21—that regulate gastrointestinal hormones align closely with clinically effective points for CAG (Table 6). Research indicates that back-shu points correspond to specific neural segments and help regulate visceral function through neuro-endocrine-immune pathways. EA at BL21 and CV12 enhances the expression of gastrointestinal hormone receptors in both the hypothalamus and gastric antrum [56]. Different acupuncture methods modulate distinct hormonal pathways: manual acupuncture primarily influences GAS and EGF, whereas moxibustion shows greater effects on MOT and SS. However, due to multiple factors influencing the effects of acupuncture, these patterns require further validation through standardized studies.

**Table 2 diseases-13-00363-t002:** Acupuncture Regulates Gastrointestinal Hormones in the Treatment of CG.

Disease	Animals	Intervention		Acupoint	Course of Treatment	Molecular Mechanism		References
CG	Wistar rats	EA	Sawtooth wave, frequency 2 Hz, voltage 2 V, current 1 mA	ST36	20 min QD d1–7		↑:GAS	[57]
CAG	SD rats	AC		BL20,BL21,BL23,BL18,BL17	15 min QD d1–30	↓:MOT	↑:GAS, EGF	[43]
			acupuncture needle 0.30 mm × 25 mm	BL17,BL18,BL20,BL21,BL23	15 min QD d1–30	↓:MOT	↑:GAS	[58]
	Wistar rats		acupuncture needle 0.25 mm × 25 mm	ST36,CV12	30 min QD d1–6, 7 d cycle, repeat × 8		↑:PGI, PGR, G-17	[53]
	SD rats	AA		ST36,ST21	1 h QD × 8 weeks	↓:CCK	↑:SP, ghrelin	[51]
	Wistar rats	Moxibustion and Chinese herb		ST36	20 min QD d1–30		↑:Gas	[59]
	SD rats			ST36,CV12	QD d1–30	↓:MOT		[60]
CNAG		Moxibustion		CV12	40 min QD d1–28	↓:MOT, IL—6	↑:GAS, SS	[48]

Disease Corresponding Abbreviations: Chronic Gastritis (CG), Chronic Non-Atrophic Gastritis (CNAG), Chronic Atrophic Gastritis (CAG).

### 3.3. Regulation of Metabolism and Gut Microbiota

Metabolic profiling reveals significant alterations in cellular processes during disease states. Acupuncture demonstrates regulatory effects on multiple metabolic pathways in CG, particularly CAG. Research examines how acupuncture modulates glucose metabolism, gastric tissue metabolism, hepatic-renal function, regional metabolism at acupoints, and gut microbiota. The stomach has been confirmed as the primary target organ in CAG. Analytical data demonstrate consistent patterns in how acupuncture, EA, and moxibustion regulate energy metabolites (including adenosine triphosphate [ATP], adenosine diphosphate [ADP], inositol, and phosphatidylcholine), amino acids, and antioxidant metabolism (Table 3, Appendix A). We systematically searched seven combinations that align with clinical research findings; therefore, they are not repeated here.

During the progression of CG, glucose, amino acid, and lipid metabolism frequently become dysregulated. Research indicated that EA at specific acupoints elicits significant metabolic changes in models of CAG. Serum metabolomics revealed that EA at ST2, ST21, and ST36 partially restored myo-inositol, very-low-density lipoprotein, and β-glucose levels in model rats [61]. These findings suggested that abnormal inositol, β-glucose, and EGF expression in CAG may reflect disturbances in energy metabolism that EA can partially correct. Liu Caichun’s comparative study demonstrated that moxibustion more effectively improved lactate and acetoacetate levels than EA [62], indicating that moxibustion may have a potentially superior regulatory effect on energy metabolism in CAG.

A comparative analysis of metabolic regulation in gastric tissue revealed distinct mechanisms underlying the effects of acupuncture and moxibustion interventions. EA modulated neurotransmitters within the nervous system, particularly affecting ethanolamine and glutamine levels, and demonstrated stronger effects on membrane metabolism than moxibustion [63]. While moxibustion primarily acted through blood-mediated energy metabolism regulation in CAG rats, EA mainly exerted therapeutic effects by repairing neural pathways in both the gastric and brain systems. This distinction aligned with previous findings: moxibustion’s warming effect promoted gastrointestinal circulation and motility, whereas acupuncture’s efficacy involved central nervous system mediation. Metabolomic studies revealed a relative specificity between stomach meridian acupoints and gastric function, with differential metabolites including glutathione, N-acetyl aspartate, phosphocholine, and uracil potentially mediating the effects of moxibustion [64]. Li Qi et al. demonstrated that moxibustion at CV12 and CV6 significantly improved antral morphological lesions, facilitated mucosal repair, and inhibited abnormal glycolytic metabolism mediated by HIF-1α [65]. Furthermore, researchers identified two key enzymes regulating glycolysis. Both warm-needle moxibustion and conventional moxibustion effectively corrected neurotransmitter and energy metabolism disorders in local acupoint tissues of CAG rats. Warm-needle moxibustion primarily modulated nucleic acid and energy metabolism, with adenosine monophosphate, hypoxanthine, inosine, and phosphocholine representing key metabolic mediators of its therapeutic effects. Moxibustion also ameliorated metabolic abnormalities by regulating multiple pathways, including alanine, aspartate, glutamate, purine, D-glutamine, and D-glutamate metabolism [66,67].

EA and moxibustion modulated metabolite alterations in liver and kidney samples from CAG models through several mechanisms [68]: (1) Energy metabolism: Moxibustion demonstrated superior regulatory effects on energy metabolism compared to EA. Both interventions normalized hepatic glycogen, glucaric acid, and glycerol levels, as well as renal glycerol phosphate, choline, glycogen, and succinic acid. However, moxibustion uniquely regulates hepatic myo-inositol, citrate, hypoxanthine, adenosine, and lysine, as well as renal nicotinamide and adenosine. (2) Neurotransmitter metabolism: EA exhibited stronger modulation of neurotransmitters in the nervous system. It restored asparagine and phenylalanine levels to normal, supporting neural function. (3) Antioxidant metabolism: Both therapies normalized glutathione and glycine levels, potentially alleviating oxidative stress via NMDA receptor activation in renal tissue. (4) Other metabolites: Both treatments significantly increased renal ethanolamine and inosine levels. Ethanolamine supports fat digestion as a component of lecithin, while inosine acts as a biomarker for renal injury. These findings suggested that acupuncture improved CAG by enhancing fat digestion and renal function.

The gut microbiota represents the body’s largest and most complex micro-ecosystem, hosting over 1000 species and functioning as an acquired “organ” [69]. In healthy individuals, diverse microbial populations maintain specific distributions along the intestinal mucosa. These communities play vital roles in sustaining normal physiological functions. Gut microbiota imbalance constitutes a significant pathological factor in the development of CAG. 16S rDNA sequencing analysis demonstrated that EA treatment altered gut microbiota composition in CAG models. Compared to model controls, EA groups showed reduced abundance of harmful bacteria, including Desulfovibrio and *H. pylori.* Conversely, beneficial bacteria such as Vibrio and Christensenellaceae exhibited increased abundance. These modifications suggested that EA might facilitate gastric mucosal repair through modulation of the gut microbiota [70].

**Table 3 diseases-13-00363-t003:** Acupuncture Regulates Metabolism and Gut Microbiota in the Treatment of CG.

Objective	Disease	Animals	Intervention		Acupoint	Course of Treatment	Molecular Mechanism		References
Regulating glucose metabolism	CAG	Wistar rats	Moxibustion		CV12, CV6	2 moxa cones/point, QD, 6 times/week × 4 weeks	↓: STAT3, HIF-1α, PKM2, LDH		[65]
Regulating tissue metabolism in acupoint areas		SD rats			ST36, CV12	15 min QD d1–14	↓: Ala, Glu, Gln, NAA, Asn, DM), Thr, Suc, PC, GPC, UDG, AMP, ADP, ATP, HX, Ino	↑: Ace, Ade, ADP	[66]
			WA		CV12, ST36	20 min QD d1–14	↓: Betaine, Threonine, Phosphocholine, Glycerophosphocholine, Adenosine Diphosphate, Inosine	↑: Lactic acid, N,N-dimethylglycine, inositol, adenosine monophosphate, adenosine, hypoxanthine	[67]
Regulating gastric tissue metabolism			Moxibustion		ST36, CV12	15 min QD × 2 weeks	↓: Leucine, Valine, N-Acetylaspartic Acid, Glutathione, Serine	↑: Glutamine, Inositol, Adenosine Ribonucleotide, Phosphocholine, Uracil	[64]
			Moxibustion/AC	Moxa sticks (l.8 cm diameter), 0.2 mm × 0.25 mm stainless steel acupuncture needles	ST36, CV12	15 min QD × 2 weeks		↑: Moxibustion: Adenosine, lactic acid, glycerol, alanine, and NADP+ levels; EGF, EGFR, and ERK Acupuncture: Adenosine monophosphate and glycerol levels; ERK	[63]
Regulating liver and kidney metabolism			Moxibustion/EA	two-channel electrical stimulations at irregular waves (intermittent wave: 4 Hz; irregular wave: 50 Hz) with voltage (2~4 V) was used.	ST36, ST21	30 min QD × 2 weeks		↑: SP, auxin-releasing peptide	[68]
Regulate fluid balance (urine and serum) and tissue metabolism (stomach, cortex, and medulla)				two-channel electrical stimulations at irregular waves (intermittent wave: 4 Hz; irregular wave: 50 Hz) with voltage (2~4 V)	ST36, ST21	30 min QD × 2 weeks	↓: glycogen, glucose and acetoacetate, glutathione and glutamine, inosine, methylmalonate and malonic acid, hypoxanthine, nicotinamide and glycerol occurred	↑: P and ghrelin, methionine, lactate and betaine, ethanolamine, phenylalanine and inositol	[62]
Regulating the Gut Microbiome			EA	were sparse and dense waves (sparse wave 4 Hz, dense wave 50 Hz) and voltage (2–4 V)	ST36	30 min QD × 4 weeks	↓: p53, c-myc, Desulfobacterota, Helicobacter	↑: Bcl-2,Oscillospirales, Romboutsia, Christensenellaceae	[70]

Disease Corresponding Abbreviations: Chronic Atrophic Gastritis (CAG).

### 3.4. Regulation of Apoptosis and Proliferation

Under physiological conditions, gastric epithelial cells maintain a balance between proliferation and apoptosis. CG progression disrupts this equilibrium, particularly evident in CAG and precancerous lesions. Early CG is characterized by excessive parietal cell apoptosis, which decreases as atrophy progresses and intestinal metaplasia develops. During this progression, epithelial cell proliferation typically increases. Research confirms that regulating cell proliferation and apoptosis represents a key mechanism of acupuncture treatment for CAG and gastric precancerous lesions (Table 4). Acupuncture and moxibustion interventions modulate the proliferation-apoptosis balance through multiple pathways. Enhancing the gastric mucosal environment effectively alleviates mucosal atrophy, regulates epithelial cell integrity, and strengthens gastric mucosal protection (Appendix A).

In precancerous or cancer transition phases, abnormal cell proliferation accelerates disease progression. Improved blood circulation in advanced chronic gastritis may also promote intestinal metaplasia and carcinogenesis. Acupuncture inhibited cell proliferation and angiogenesis through multiple mechanisms, ultimately improving gastric mucosal morphology in models of chronic atrophic gastritis. Specific mechanisms included: (1) Regulation of oncogenic and proliferative markers: The oncogene c-myc disrupts cellular metabolism when overexpressed [71,72]. Proliferating cell nuclear antigen (PCNA) serves as an established marker of cell proliferation and plays crucial roles in DNA replication and repair [73,74]. (2) Modulation of growth factor signaling: EGF drives cell proliferation, with studies demonstrating that aberrant EGF signaling promoted gastrointestinal carcinogenesis [75]. Transforming Growth Factor-α (TGF-α) represents another oncogenic factor whose upregulation correlated with cancer cell proliferation [76]. VEGF stimulates angiogenesis by enhancing vascular permeability and endothelial cell nutrition. (3) Control of Hedgehog pathway effectors: Gli proteins (Gli1/2/3) function as terminal effectors in the Hedgehog signaling pathway, participating in endothelial cell proliferation and angiogenesis processes [72,77].

Moxibustion modulated the proliferation-apoptosis balance in gastric mucosal cells of chronic gastritis patients. Zhang Haifeng et al. [35] demonstrated that heat-sensitive moxibustion at CV12 downregulated c-myc, Cyclin D1, and survivin expression. This regulation involved cell cycle control and rebalanced proliferation-apoptosis dynamics, improving the gastric mucosal microenvironment. Moxibustion upregulated key gastric mucins, including MUC1, MUC5AC, and MUC6, thereby enhancing gastroduodenal epithelial integrity while suppressing excessive proliferation [29]. Cai Jin et al. confirmed moxibustion modulated the TGF-β/Smad pathway, boosting endogenous protective factors and mitigating proliferative abnormalities in CAG precancerous models [78].

Additional regulatory pathways included actin-binding protein and Notch signaling. Actin-binding protein mediates cytoskeletal reorganization in tumor development, while Notch signaling coordinates cell differentiation and proliferation. Wang Shuguo et al. [79] demonstrated that electroacupuncture activates the Fas/FasL pathway in mice with chronic atrophic gastritis, thereby ameliorating gastric mucosal atrophy. Emerging evidence suggested that catgut embedding and acupoint application similarly regulate the proliferation-apoptosis equilibrium. Wang Kun et al. [80] found that catgut embedding upregulated SOCS3, thereby inhibiting JAK2-STAT3 signaling and downregulating Bcl-2 and Cyclin D1, which contributes to gastric mucosal protection. Acupoint application downregulated TNF-α and PCNA expression, thereby reducing gastric mucosal inflammation and inhibiting abnormal proliferation in rats with chronic atrophic gastritis [81].

**Table 4 diseases-13-00363-t004:** Acupuncture Regulates Apoptosis and Proliferation in the Treatment of CG.

Disease	Animals	Intervention		Acupoint	Course of Treatment	Molecular Mechanism		References
CAG	Wistar rats	Moxibustion		CV12,CV6	2 moxa cones/acupoint, QD, 6 times/week × 4 weeks	↓:P-AKT,PIP2,MDM2	↑:PTEN/Caspase-9, PI3KCA	[82]
				CV12,CV6	1 moxa cones/acupoint, QD × 4 weeks		↑:Foxo3,Uba52,S100a1,Nod2	[83]
	SD rats			CV12,CV6	2 moxa cones/acupoint, QD, 6 times/week × 4 weeks	↓:STAT3,HIF-1α,PKM2,LDH		[65]
				CV12	40 min QD d1–28	↓:cmyc, survivin, cyclin D1		[71]
PLGC				ST36,ST21	30 min QD × 20 weeks	↓:EGF, TGF-a, PCNA, VEGF, Ag-NORs		[72]
	Wistar rats			CV12,ST36	40 min QD × 4 weeks	↓:EGF, TGF-β, P53, Bcl-2, Ag-NORs, PCNA	↑: Smad3	[78]
		Moxibustion and Chinese herb		BL18,BL20,BL21,ST36,PC6,CV12	1 moxa cone/acupoint,QD × 10 d,then rest 2–3 d. repeat × 3.	↓:Survivin, p53	¬:Syk	[84]
CAG	SD rats	Moxibustion/AC	Moxibustion stick 1.8 × 12 cm, Ac needles SS 0.25 × 25 mm	ST36,CV12	15 min QD × 2 weeks	↓:NF-KB, Bcl-2		[85]
		AC	DS Ac needles 0.3 × 40 mm	CV12,PC6,ST36	3 times/week, 10 sessions/course, ×6 courses		↑: Notch2, Notch3	[86]
		ACE	4-0 Catgut Suture, 0.5 cm Needle, 6-Gauge Injection Needle	BL20,ST36	Q10D × 6	↓:JAK2, STAT3, Bcl-2, CyclinD1	↑:SOCS3	[80]
		EA	Disperse-Dense wave (4/20 Hz, 60 V)	ST36,ST21	30 min QD × 4 weeks	↓:P53		[87]
			Electro-acupuncture: Intermittent–Irregular wave (4/50 Hz, 2–4 V)	ST2,ST36,ST21	30 min QD × 2 weeks	↓: PCNA, Ag-NORs, EGF, VEGF, c-myc, NF-κB		[60]

Disease Corresponding Abbreviations: Chronic Atrophic Gastritis (CAG), Precancerous Lesions of Gastric Cancer (PLGC).

### 3.5. Anti-Inflammatory and Antioxidant Stress

Persistent gastric mucosal inflammation drives CG progression to gastric cancer. Chronic inflammation independently promotes gastric carcinogenesis, regardless of *H. pylori* infection status [88]. Nonsteroidal anti-inflammatory drugs effectively prevent gastric precancerous lesions but cause adverse effects, including abdominal pain and gastrointestinal bleeding [89]. This underscores the need for safer anti-inflammatory strategies for CG. Acupuncture demonstrates anti-inflammatory effects in both experimental models and clinical practice. These benefits extend to inflammatory bowel disease, knee osteoarthritis, and Crohn’s disease [90,91,92,93]. Acupuncture and related therapies alleviate gastric inflammation primarily by inhibiting pro-inflammatory cytokines and lipids while activating protective signaling pathways (Appendix A, Table 5).

Pro-inflammatory cytokines play central roles in gastric inflammation. Acupuncture significantly reduced TNF-α, IL-1β, IL-2, IL-4, IL-6, IL-8, and IL-10 expression in chronic gastritis [CG]. COX-2 and PGE2 also contribute to the initiation of inflammation. COX-2 overexpression accelerates inflammatory factor release [94], while PGE2 modulates macrophages and lymphocytes at inflammatory sites [95]. Chronic atrophic gastritis [CAG] models exhibited elevated COX-2 but reduced PGE2 and PGF2α in the gastric mucosa. Acupuncture normalized these protein levels and alleviated inflammation [31,77]. Epithelial–mesenchymal transition generates pro-inflammatory factors in cancer progression [96]. Cytokeratin serves as a biomarker for epithelial–mesenchymal transition in studies of chronic inflammation and cancer [97,98]. Recent research implicates neuropeptide Y and calcitonin gene-related peptide in inflammatory processes [99,100]. Combined acupuncture-moxibustion at ST36 enhanced CK18, CK19, and calcitonin gene-related peptide expression while reducing neuropeptide Y, ultimately mitigating inflammation [101].

Research on acupoint sensitization reveals that visceral inflammation alters the local microenvironment of acupoints. These changes include increased expression of histamine and substance P [102]. Studies have demonstrated that acupuncture modulates substance P and histamine receptor H2 expression at acupoints in rats with chronic atrophic gastritis, suggesting that this mechanism contributes to its therapeutic effects [103]. Acupuncture’s anti-inflammatory action also involves regulating signaling pathways, including NF-κB, its p65 subunit, and the Fas/FasL system [79,93,104]. NF-κB pathway activation promotes gastric mucosal inflammation, while ubiquitin-mediated regulation of NF-κB signaling exacerbates inflammatory responses [105]. The Fas/FasL system regulates both apoptosis and inflammation, with abnormal signaling contributing to the formation of an inflammatory microenvironment [106].

Oxidative stress represents an imbalance between oxidative and antioxidant systems. Acupuncture and catgut embedding regulated superoxide dismutase [SOD] and malondialdehyde [MDA] levels in chronic atrophic gastritis [107,108]. SOD catalyzes the neutralization of free radicals to reduce cellular damage. MDA indicates lipid peroxidation extent from oxidative injury. Combined SOD deficiency and MDA elevation promoted gastric mucosal damage [109].

**Table 5 diseases-13-00363-t005:** Acupuncture for Anti-inflammatory and Antioxidant Stress Treatment in CG.

Objective	Disease	Animals	Intervention		Acupoint	Course of Treatment	Molecular Mechanism		References
Anti-inflammatory	CAG	SD rats	Moxibustion	Moxa stick 8 mm × 9 mm	ST36,ST25	20 min QD × 2 weeks	↓: COX-2,NF-κBp65	↑: miR-146a	[110]
				Moxa stick, 1.8 cm diameter	ST36	15 min QD d1–14		↑: IL-1β,IL-10,TNF-α	[36]
			Moxibustion and AC	Moxa stick 0.8 cm × 12 cm	ST36,CV12	15 min QD d1–14	↓: CK18,CK19	↑: CGRP, NPY	[101]
		Wistar rats	AC		ST36,SP6,BL25,CV6,CV12,CV10,ST25,LI10, BL26,BL20,SP10	15 min QOD × 4 weeks	↓: GLi1,GLi2,GLi3,COX-2,TNF-α,G-17,IL-1β,IL-4	:	[77]
		SD rats		0.25 mm × 25 mm stainless steel acupuncture needles	ST36	20 min QD d1–14	↓: IL-6,IL-1β,TNF-α	↑: SP, HRH2	[103]
					ST36,CV12	15 min QD d1–60	↓:miR-155, miR-21,NF-κB p65	↑: miR-146a	[111]
			ACE	0.5 cm 0000 catgut suture 6-gauge injection needle	BL20,ST36,CV12	q10d × 6	↓: IL-1β,IL-6,TNF-α		[112]
			AA		ST36,ST21	1 h QD × 8 weeks	↓: TNF-α, PCNA		[81]
			EA	intermittent and irregular waves (intermittent wave: 4 Hz; irregular wave: 50 Hz) with voltage ranging from 2 to 4 V.	ST2,ST36,ST21	30 min QD × 2 weeks	↓: PCNA, Ag-NORs, EGF, VEGF, c-myc, NF-κB		[61]
		C57BL/6 mouse			ST36,CV12	30 min QD d1–14	↓: IL-6, IL-1β, Fas, FasL		[79]
	CG	Wistar rats	EA and Chinese herb	EA 2 Hz, 3 V, 0.3 mA	BL20,BL21	15 min QOD × 6 weeks	↓: IL-10,TNF-α		[113]
	CAG		LA		ST36	6 min QD d1–14	↓: IL-2,IL-6,TGF-β1		[114]
			ear acupoint bean pressing and Chinese herb		CO13,CO4,AH6a,CO18,AT4	10 min TID, alternate ears, change q3d × 6 months	↓: NF-κB, Ubiquitin		[115]
Antioxidant Stress		SD rats	ACE	4-0 chromic catgut suture, 0.5 cm in length, Size 6 disposable injection needle	BL20,CV12,ST36	q10d × 6	↓: MDA	↑: SOD	[106]
			AC		BL17,BL18,BL20,BL21,BL23	15 min QD d1–30	↓:MDA	↑:SOD	[107]

Disease Corresponding Abbreviations: Chronic Gastritis (CG), Chronic Atrophic Gastritis (CAG).

## 4. Current Status of Clinical Research on Acupuncture Treatment for CG

CG involves complex mechanisms and presents with symptoms such as bloating, discomfort, acid reflux, and reduced appetite. This common chronic condition substantially affects quality of life [116]. Mendelian randomization indicates that genetic depression predisposition correlates with CG development [117]. Western medicine manages symptoms using antibiotics, proton pump inhibitors, bismuth agents, and acid suppressants. These treatments control symptoms effectively but face challenges, including prolonged duration, frequent recurrence, and gastrointestinal side effects [118]. TCM provides personalized treatments that enable comprehensive physiological regulation. This review systematically evaluates the progress of acupuncture research in CG over the past decade. It categorizes interventions including acupuncture, moxibustion, acupoint application, and combination therapies. The analysis aims to provide valuable insights for the treatment and research of CG.

### 4.1. Acupuncture and EA

Acupuncture and EA offer distinct therapeutic benefits for CG by targeting specific acupoints, such as ST36, CV12, and PC6 (Table 6). These interventions enhance gastrointestinal motility, reduce inflammation, and promote mucosal healing. Small-scale clinical trials demonstrated that EA achieved a 90.9% response rate in CNAG, potentially improving serum GAS levels and electrogastrogram parameters through neuroendocrine regulation [119]. Multiple studies have reported reduced inflammatory factors alongside increased CD4^+^ and CD8^+^ cells, indicating that acupuncture alleviates inflammation and enhances immune function [120,121,122]. For CAG, combining acupuncture with specialized needle techniques relieved symptoms and regulated gastrointestinal hormones, including GAS, MOT, and SS [123,124]. While some studies have observed reduced glandular atrophy and intestinal metaplasia, these findings require validation through multicenter, randomized trials with long-term follow-up [125]. Preliminary mechanistic studies suggested that acupuncture regulated actin-binding proteins and Notch pathway components, though this evidence required further confirmation [86].

### 4.2. Moxibustion

Moxibustion applies heat to specific acupoints through burning moxa. This TCM therapy is effective in treating CG, particularly at the points CV12, ST36, CV6, and PC6 (Table 6). Research has demonstrated that moxibustion warms meridians, modulates immune function, and improves syndrome scores and quality of life (as measured by the SF-36). Although minor adverse events, such as local burns, occasionally occurred, most studies confirmed a manageable safety profile [126,127,128]. Several factors influenced the efficacy of moxibustion, including technique selection, acupoint location, session duration, and treatment course. Clinical comparisons have shown that sensitized acupoints yield better outcomes than non-sensitized points in CAG [129]. Various techniques—including ginger-separated, mu-shu combined, and coiled moxa—improved gastric mucosal pathology. These methods elevated serum pepsinogen (PGⅠ, PGⅡ) and G-17 levels, suggesting enhanced mucosal recovery [130,131,132]. Mechanistic studies suggest that moxibustion suppresses inflammation and regulates energy metabolism through the cAMP, AMPK, and NF-κB pathways. It also influenced gastric repair through epigenetic mechanisms, such as DNA methylation [133]. Furthermore, moxibustion regulates brain–gut peptides (ghrelin, MOT, SS), providing insights into central-peripheral integration mechanisms [134].

### 4.3. Warm Needle Acupuncture (WA)

WA combines needling with thermal stimulation, promoting vasodilation and local metabolism. This therapy shows positive outcomes for CAG and CNAG when applied at CV12, ST36, and CV4 (Table 6). Clinical studies have indicated that WA outperforms conventional acupuncture for CNAG, demonstrating greater improvements in TCM syndrome scores and quality of life [135,136,137]. Gastroscopic and histopathological analyses further associated WA’s symptom relief with enhanced gastric mucosal health [138]. However, high-quality objective evidence remained limited for CAG applications. Reports of variable patient satisfaction further highlighted inconsistent therapeutic benefits. Thus, current evidence remained insufficient to establish WA’s efficacy for CAG [139].

### 4.4. Other Therapies

Alternative acupuncture methods, such as acupoint application, thread implantation, and injection, improve gastric mucosal health and alleviate symptoms. Commonly used acupoints include CV12, ST36, and BL20 (Table 6). Acupoint application therapy combined acupoint stimulation with herbal effects to regulate GAS and enhance gastric function, producing positive outcomes for digestive disorders and chronic pain [140]. One study using Astragalus-Lotus Seed paste reported a 66% *H. pylori* eradication rate in CAG, although larger trials were needed for validation [141]. Visual Analogue Scale assessments demonstrated that acupoint plaster effectively reduced epigastric pain with few adverse reactions, potentially complementing Western medications [142,143]. Minor skin reactions occasionally occur but can be minimized through formula adjustments [144]. Thread implantation provided sustained biochemical stimulation, improving clinical symptoms and mucosal morphology in CAG patients. This therapy reduced TCM syndrome scores, possibly through the regulation of gastrointestinal function and hormone modulation [145]. Acupoint injection delivers medications directly to acupoints, enhancing mucosal blood flow and potentially aiding in the treatment of chronic gastritis through localized, sustained stimulation [146].

**Table 6 diseases-13-00363-t006:** Acupuncture (including EA), moxibustion, warm needle moxibustion, and other monotherapies for treating CG.

Disease	Location	Center	Age (Mean ± SD), y	Intervention	Acupoint	Time of Intervention	Duration	N	Control			Measurement Time Points	Validated Scale Used	Objective Evaluation Criteria	Post-Acupuncture Changes Compared to Baseline	Adverse Events	
									Age (Mean ± SD), y	Treatment	N						References
CNAG	CN	SC	39 ± 11	EA	ST2,ST21,ST36	30 min QD	4 weeks	33	38 ± 11	Conv acup	33	Before treatment, after treatment	TCM Consensus on Chronic Superficial Gastritis Dx & Tx, Chinese Consensus on Chronic Gastritis Symptom Score	S-GAS levels, gastrogram parameters	Total effective rate 90.9%; total scores decreased, S-GAS decreased; gastric electrical activity amplitude and frequency increased, rhythm disorders decreased (*p* < 0.05)		[119]
CNAG	CN	SC	42 ± 5	AC	ST36, RN12,SP3, ST40	AC: QD, 5 times/week, needles retained 30 min, manipulation every 10 min	4 weeks	37	41 ± 5	Conv acup	37	Before treatment, after treatment	Primary symptom score, Secondary symptom score	Inflammatory factors, gastric mucosal repair indicators	Total effective rate 83.8% (*p* < 0.05); primary and secondary symptom scores decreased, serum inflammatory factors decreased, gastric mucosal repair indicators improved (*p* < 0.05)		[121]
CNAG	CN	SC	41.5 ± 9.4	AC	ST36,RN12,RN4,ST25,DU24,EX-HN1,PC6,SP6	QOD, TIW, needles retained 30 min	8 weeks	30	42.3 ± 8.7	Oral omeprazole + sucralfate susp + mosapride citrate PRN, per Rx for 8 weeks	30	Before treatment, after treatment	Guidelines for New TCM Drugs Clinical Research, GI Diseases TCMSS, SAS, SDS, etc.	Gastroscopy score	Total effective rate 86.67%; GS, symptom scores, SAS, SDS decreased (*p* < 0.05)		[147]
CAG	CN	SC	51 ± 7	AC	RN12,ST36,PC6,SP4,RN4,BL20,BL21	QD, needles retained 30 min, manipulation every 15 min	2.14 weeks	31	51 ± 6	Conv acup	31	Before treatment, after treatment	TCM Consensus on Chronic Atrophic Gastritis (2009), Guidelines for New TCM Drugs Clinical Research-TCMSS		Total effective rate 93.5% (*p* < 0.05); TCMSS decreased (*p* < 0.05)		[148]
CAG	CN	SC	46.28 ± 1.31	Ten Old Needles	ST25,PC6,BL21,RN13,RN12,RN10,RN6	QD, 5 times/week, needles retained 20 min	8 weeks	25	45.96 ± 1.54	Oral vit tabs + metoclopramide tabs + rabeprazole EC caps, per Rx for 8 weeks	25	Before treatment, after treatment	Chinese Consensus on Chronic Gastritis, Guidelines for New TCM Drugs Clinical Research—TCMSS	Gastric mucosal gland atrophy degree, intestinal metaplasia degree	Total effective rate 92% (*p* < 0.05); gastric mucosal gland atrophy, intestinal metaplasia, TCMSS decreased (*p* < 0.05)		[124]
CAG	CN	SC	55.36 ± 5.35	AC + FSN	ST40,SP3,SP4,ST42	Tech1: QD, 5 times/week; Tech2: QOD, TIW	4 weeks	50	55.26 ± 5.29	FSN ther	56	Before treatment, after treatment	TCMSSS, GSRS, SF-36 PCS	Gastric mucosal score, S-GAS, S-MOT, S-SS levels	Total effective rate 96.00% (*p* < 0.05); TCM syndrome scores decreased, PCS increased, S-GAS, S-MOT, S-SS decreased, gastric mucosal scores decreased (*p* < 0.05)		[123]
CAG	CN	SC	56.43 ± 5.80	AC + Fire needle	RN12,BL21,SP4,ST42	Needle1: QOD, TIW; Tech: QD, 5 times/week	4 weeks	49	56.55 ± 5.88	Filiform Fire Needle	49	Before treatment, after treatment	TCMSSS, SF-36 PCS	Gastric mucosal score, PCS score, S-GAS/MOT/SS	Total effective rate 95.92% (*p* < 0.05); TCM syndrome scores decreased, PCS increased, gastric mucosal scores decreased, S-GAS and S-MOT decreased, S-SS increased (*p* < 0.05)		[124]
CAG	Chinese	SC	52.32 ± 1.75	AC	ST36,RN12,BL20,LI4,KI3	needles retained 30 min, manipulation every 10 min, 5 times/week	12 weeks	53	54.26 ± 1.65	Oral omeprazole + amoxicillin + metronidazole × 2 wks + celecoxib until 12 wks	53	Before treatment, 2 weeks after treatment, 12 weeks after treatment	No specific scale	Gastroscopy findings, *H. pylori* eradication rate	Total effective rate 92.5%; endoscopic efficacy 88.7% (*p* < 0.05); *H. pylori* eradication rate 84.9%		[149]
CAG	CN	SC	58 ± 8	AC	RN13, RN12	2 times/week, needles retained 30 min with techniques	24 weeks	36	59 ± 8	Conv acup	35	Before treatment, after treatment, 6-month follow-up	TCMSS	Gastroscopic mucosal scoring	Total effective rate 86.1% (*p* < 0.05); TCMSS and gastroscopic mucosal scores decreased		[150]
CAG	CN	SC	62.63 ± 3.66	AC	RN12, PC6,ST36	TIW, needles retained 20 min, 10 sessions/course	15–16 weeks	8		Untreated	Each group consists of 8 people.	Before treatment, after treatment	TCMSSS	Serum protein testing	Protein levels: thymosin β-4, Profilin-1, myosin-4, transglutaminase-2 decreased; Notch2, Notch3 increased (*p* < 0.05)		[86]
CG	CN	SC	54.4 ± 8.5	AC	DU24,DU22,DU21,PC6,ST36,RN12	30 min QD, manipulation every 10 min	4.57 weeks	46	54.8 ± 8.4	Conv acup	46	Before treatment, after treatment	Chinese Consensus on Chronic Gastritis, ICD-10, SAS, SDS, etc.	Clinical symptom score, gastroscopic morphology, SAS/SDS score, clinical efficacy	Clinical symptom scores decreased, gastroscopic improvements, SAS and SDS decreased (*p* < 0.05)		[151]
CG	CN	SC	43.2	AC	LI4,LR3,RN12	10 min/session	Single treatment	40		Oral scopolamine tabs 20 mg/dose	40	Before treatment, 30 min after treatment, 60 min after treatment		Gastroscopy/barium meal follow-up results	Total effective rate 95% (*p* < 0.05); faster and more pronounced pain relief		[152]
CG	CN	SC	47.89 ± 7.03	AC	Primary Point: Taiyin of the Head. Additional Points Based on Pattern Differentiation: For Liver and Stomach Qi Stagnation, add Taiyang / Jueyin; For Liver and Stomach Heat Stagnation, add Yangming / Jueyin.	30 min QD, 7-day course	1 week	35	48.96 ± 7.14	Oral omeprazole 10 mg QD + mosapride citrate 5 mg TID	35	Before treatment, after treatment	TCMSS System, TCM Expert Consensus on Functional Dyspepsia (2017)—upper abdominal pain syndrome score	GI hormones, pepsin, gastroscopy/pathology score	Total effective rate 94.29%; TCMSS, upper abdominal pain, upper abdominal burning, gastroscopy, pathology decreased (*p* < 0.05); S-G-17, S-MOT, S-PGI, S-PGII increased (*p* < 0.05); recurrence rate 8.57%	Adverse reactions occurred in 6 individuals.	[120]
CG	CN	SC	46.83 ± 10.75	AC	EX-B3, BL20, BL21, BL18, BL19	QD, 5 times/week, needles retained after deqi	4 weeks	30	46.83 ± 11.03	Conv western meds for triple therapy for gastritis	30	Before treatment, after treatment	TCMSS	S-CD_4_^+^, S-CD_8_^+^, S-CD_4_^+^/CD_8_^+^ ratio	Improvement rate 70% (*p* < 0.05); TCMSS decreased, S-CD4^+^ and S-CD4^+^/CD8^+^ ratios increased (*p* < 0.05)		[122]
CNAG	CN	SC	52.22 + 9.70	Mox	CV12, PC, ST36	QD, 30 min per session	8 weeks	29	63.54 + 8.97	Mox at non-sensitive points	20	Before treatment, After treatment, Follow-up	TCMSSS, SF-36		Total effective rate 100%; TCMSS decreased (*p* < 0.05), SF-36 scores increased (*p* < 0.05)		[129]
CNAG	SC		46.02 ± 10.12	Mox	From the xiphoid process to the umbilicus, with lateral margins extending to the midclavicular line.	Each session: wait until moxa wool self-ignites before replacing, repeat for 3 consecutive times, total about 60 min, once every 3 days	3 weeks	25	44.83 ± 9.54	Oral lansoprazole EC tabs 30 mg QD	25	Before treatment, After treatment	“GI Diseases TCM Syndrome Rating Scale”		Total effective rate 92%; TCMSS decreased (*p* < 0.05)		[132]
CAG	CN	SC	57.63 ± 9.17	Electronic Mox	ST36,CV12,ST25,CV8	QD, 30 min per session	2 weeks	30	52.06 ± 12.61	Conv treat	30	Before treatment, Day 7 of treatment, Day 14 of treatment	TCMSSS, VAS		Total effective rate 93%; TCMSS and pain scores decreased (*p* < 0.05)		[127]
CAG	CN	SC	61.55	Mox	CV12,CV6,ST36,PC6	1 unit per point per session, TIW	24 weeks	33	59.36	Mox	33	Before treatment, After treatment	“GI Diseases TCM Symptom Rating Scale”	Gastroscopy: gastric mucosa histopathology; peripheral blood DNA methylation sequencing	Total effective rate 87.88%; gastric mucosa pathological efficacy 82.14%; TCMSSS, individual TCMSS, gastric mucosa tissue lesion scores decreased (*p* < 0.05); post-treatment DMR genes involved cAMP, AMPK, NF-κB pathways		[133]
CAG	CN	SC	59.90 ± 10.5	Mox	CV12,CV4,PC6,ST36	30 min per session, QOD, TIW	4 weeks	40	60.60 ± 9.56	Smokeless mox	40	Before treatment, After treatment	“GI Diseases TCM Syndrome Rating Scale”	Acupoint temperature, S-GAS, S-PGI, S-PGR	Total effective rate 75%; TCMSS decreased (*p* < 0.05); S-GAS, S-PGI, S-PGR increased (*p* < 0.05)		[153]
CAG	CN	SC		Mox	CV12,CV4,PC6,ST36	1 unit per point per session	4.57 weeks	47		Western med conv treat	48	Before treatment, After treatment	TCMSSS	Gastroscopy: gastric mucosa histomorphology; S-G-17, S-PGR	Total effective rate 89.4%; TCMSS, gastroscopy, pathological gastric mucosa histomorphology scores decreased (*p* < 0.05); S-G-17, S-PGR increased (*p* < 0.05)	Five cases developed small blisters at the ginger-isolated Mox site, which healed completely after management with no complications such as infection.	[126]
CAG	SC		57 + 11	Mox	CV12,ST36	20–25 min per session, QD	12 weeks	32	55 ± 9	Oral Weifuchun tabs 1.44 g TID	31	Before treatment, After treatment, Follow-up	TCMSSS	S-PGI, S-PGII, S-G-17	Total effective rate 93.8%; TCMSS and gastroscopic gastric mucosa scores decreased (*p* < 0.05); S-PGI, S-PGR, S-G-17 increased (*p* < 0.01)		[131]
CAG	CN	SC	51.82 + 11.45	Mox	From GV14 to GV1	QW, for 3 consecutive sessions	12 weeks	32	52.56 + 9.62	Oral Huangqi Jianzhong decoc	32	Before treatment, Week 4, Week 8, Week 12	TCMSSS	Gastroscopy: gastric mucosa score	Total effective rate 96.67%; TCMSS and gastroscopic gastric mucosa scores decreased (*p* < 0.05)		[130]
CG	CN	SC	59.5	Mox	CV12,CV6,ST25	25 min per point, QOD, TIW	4 weeks	30	56.5	Mox	30	Before treatment, After treatment	TCMSS for Digestive Diseases, VAS	S-Ghrelin, S-SS, S-MOT	Gastrointestinal disease-related TCMSS and VAS scores decreased (*p* < 0.05); S-SS decreased (*p* < 0.05), S-Ghrelin, S-MOT increased (*p* < 0.05)	One adverse event occurred during the treatment period. This event was a mild burn during the Mox procedure, which healed before the next treatment session. No serious adverse events occurred.	[134]
CG	CN	SC	60.45 ± 10.37	Mox	CV12,CV6,PC6,ST36	1 unit per point per session, TIW	4 weeks	31	60.71 ± 10.22	Mox placebo	31	Before treatment, After treatment	TCMSSS	S-PGI, S-PGII, S-PGR, S-GAS	Total effective rate 83.9%; TCMSS and S-GAS decreased (*p* < 0.05); S-PGI, S-PGR increased (*p* < 0.05)		[128]
CNAG	CN	SC	39.6 ± 3.3	WA	SP4,PC6,SP9,RN12,ST36,RN4	2 cones/point, QOD	1.71 weeks	50	39.2 ± 3.7	Conv acup	50	before treatment, after treatment	TCM syndrome rating scale, TCMSSS, SF-36		Total effective rate 90%; TCMSS decreased (*p* < 0.05), SF-36 scores increased (*p* < 0.05)		[137]
CNAG	CN	SC	42.88 ± 3.82	WA	ST36,RN4,RN12	2 cones/point, QD	5.71 weeks	37	42.56 ± 3.75	Conv acup	35	before treatment, after treatment	TCM syndrome rating scale, TCMSSS, SF-36		Total effective rate 96.83%; TCMSS decreased (*p* < 0.05), SF-36 scores increased (*p* < 0.05)		[135]
CNAG	CN	SC	47.56 ± 2.82	WA	ST36,SP4,SP9,PC6,RN12	2 cones/point, QOD	1.71 weeks	40	46.83 ± 3.62	Conv acup	39	before treatment, after treatment	TCM syndrome rating scale, TCMSSS, SF-36		Total effective rate 62.5%; TCMSS decreased (*p* < 0.05), SF-36 scores increased (*p* < 0.05)		[136]
CNAG	CN	SC	48.08 ± 11.32	WA	DU20,EX-HN1,MS3,RN12,SP15,RN4,RN6,PC6,ST36,SP6,SP4,ST44	30 min, QD	8 weeks	43	47.99 ± 11.19	Conv Western med treat	43	before treatment, 2 weeks after treatment, 4 weeks after treatment, 6 weeks after treatment	TCM syndrome rating scale, TCMSSS, SF-36	Gastroscopy, gastric mucosal biopsy	Total effective rate 90.7%; GS decreased (*p* < 0.05); *H. pylori* eradication rate 93.02%; recurrence rate 9.3% at 1-year follow-up		[138]
CG	CN	SC	48 ± 9	WA	RN12,RN4,ST36,BL17,SP10,BL20,BL21	30 min, QD	8 weeks	30	48 ± 9	Oral omeprazole caps 20 mg QAM; amoxicillin caps 0.5 g BID; metronidazole tabs 0.4 g BID	30	before treatment, after treatment	Guidelines for New TCM Drugs Clinical Research—efficacy standards	Gastroscopy	Total effective rate 93.3%; GS decreased (*p* < 0.05)		[154]
CAG	CN	SC	52 ± 6	WA	RN4,ST36,RN17,RN8,RN12	QOD	4 weeks	63	53 ± 6	Oral rabeprazole EC tabs 10 mg BID	63	before treatment, 1 week after treatment, 2 weeks after treatment, 1 month after treatment	TCM syndrome rating scale, TCMSS		TCMSS decreased (*p* < 0.05)	Four patients were dissatisfied with the treatment.	[155]
CG	CN	SC	47.61 ± 7.65	WA Combined AC Method (AC + WA)	WA: EX-HN1, DU20, RN12, RN4, SP15, RN6, ST36, ST44, PC6, SP4, SP6;AC: ST36, Left SP3, Left LR3	WA: QOD, needles retained 30 min; Rotation: QOD, needles retained 30 min	8 weeks	60	47.69 ± 7.71	Pure WA ther (Western med)	60	Before treatment, 2 months after treatment	GSRS		Total effective rate 93.33%; GS decreased (*p* < 0.05); S-MOT and S-GAS increased, S-VIP decreased (*p* < 0.05); *H. pylori* conversion rate 79.24%		[156]
CAG	CN	SC	52.44 ± 3.2	AA	RN12,BL20,BL21	24 h, QOD	24 weeks	80	51.95 ± 3.6	Weifuchun tabs 4 tabs TID	80	Before treatment, after treatment	TCM Symptom Score	Gastroscopy: mucosal score, pathological score; 13C-UBT	Total effective rate 77.92%; *H. pylori* conversion rate 65.63%, pathological efficacy 71.43%, gastroscopic efficacy 76.63%, syndrome efficacy 89.61%; symptom scores, syndrome scores, mucosal image scores, pathological scores decreased (*p* < 0.05)	One case of localized allergy	[141]
CG	CN	SC	37.47 + 8.81	AA	RN12,RN8,ST36,BL20	6–8 h, TIW	4 weeks	30	34.53 ± 9.60	Oral omeprazole EC caps 20 mg/dose	30	Before treatment, after treatment	VAS		Total effective rate 86.7%; VAS for stomach pain decreased (*p* < 0.05)		[142]
CG	CN	SC	54.27 + 10.92	AA	RN8,RN12,BL18,BL20,BL21,ST36	4 h, QD	1 week	44	54.76 + 11.28	Placebo AA	42	Before treatment, 3 days after treatment, 7 days after treatment	TCMSS per Guidelines for New TCM Drugs Clinical Research		TCMSS decreased (*p* < 0.05)		[140]
CG	CN	SC	51.25 ± 12.59	AA	ST36,BL20,RN12,RN13,RN4,RN6	4 h, QD, d1–5	1.43 weeks	40	55.28 ± 10.56	Treat and nursing per TCM clinical pathway	40	Before treatment, 3 days after treatment, 10 days after treatment	VAS, TCMSS per Guidelines for New TCM Drugs Clinical Research		Total effective rate 97.5%; gastric pain VAS and TCMSS decreased (*p* < 0.05)	Five patients experienced itching at the AA sites, which resolved spontaneously after discontinuing the medication. One patient developed an allergic reaction at the AA site.	[143]
CG	CN	SC		AA	RN13,RN12,RN11,ST25	QD	1.43 weeks	30		Pantoprazole sodium 40 mg IV BID; oral herbal decoc 100 mL TID	30	Before treatment, after treatment	GI Diseases TCMSS		Total effective rate 96.7%; TCMSS decreased (*p* < 0.05)		[144]
CAG	CN	SC		ACE	ST36,BL20,BL21,ST37,ST39, RN12	QW	24 weeks	65			45	After treatment	GI Diseases TCMSS	Gastroscopy: gastric mucosal lesion degree, gastric mucosa histopathology			[145]
CG	CN	SC	48.72 ± 9.13	ACE	RN12,ST21,ST36,SP	QD	1 week	45	46.31 ± 8.64	Omeprazole IV 40 mg	45	Before treatment, after treatment	TCMSS per Guidelines for New TCM Drugs Clinical Research		Total effective rate 95.55%; TCMSS decreased (*p* < 0.05)		[146]

Disease Corresponding Abbreviations: Chronic Gastritis (CG), Chronic Non-Atrophic Gastritis (CNAG), Chronic Atrophic Gastritis (CAG). China: CN, Single-center: SC, QD: Once a day, BID: Twice a day, TID: Three times a day, QID: Four times a day, QOD: Every other day, Q2W: Every two weeks, TIW: Three times a week, d1-x: for x consecutive days, xd: x days, QW: Once a week.

### 4.5. Combined Therapy

Therapies that integrate acupuncture are commonly applied to address digestive system disorders. In practice, various techniques are employed, including pairing acupuncture with herbal treatments, combining moxibustion with herbal remedies, applying acupoint plasters with medication, and integrating acupuncture with thread implantation. These strategies effectively ease the symptoms associated with CG, alleviate patient discomfort by enhancing gastric function, balancing gastric hormone levels, and promoting the health of the gastric mucosa. Key acupoints often targeted in these treatments are CV12, ST36, BL20, and BL21 (Appendix A).

Combination therapy demonstrated notable efficacy for CNAG. Integrating TCM external treatments with conventional drugs provided dual therapeutic benefits [157]. Acupoint plasters combined with omeprazole significantly reduced GAS levels while increasing MOT and SS [158]. This approach alleviated symptoms and improved patient satisfaction [159]. Acupoint plasters with herbal medicine enhanced gastrointestinal hormones, improved motility, and restored function in patients with CNAG. Combining TCM with thread implantation reduced serum levels of IL-32, CGRP, and EGF, thereby diminishing inflammatory damage and recurrence rates [160]. Compared to Western monotherapy, proton pump inhibitors with thread implantation showed superior symptom relief despite similar cure rates [161]. Small-scale trials confirmed that warm acupuncture with medication significantly reduced TNF-α, IL-8, IL-1β, and hs-CRP levels in CNAG patients. This combination also enhanced H. pylori eradication rates through the combined effects of thermal stimulation and gastric protection [162]. Moxibustion with herbal medicine proved more effective than moxibustion alone for CAG. This combination improved patient condition, reduced serum IL-6, and produced significant meridian temperature changes [163]. Additionally, acupuncture with thread implantation enhanced gastric mucosal blood circulation, relieved discomfort, and improved quality of life [164].

Acupuncture combined with medication represents a widely used approach for CAG. Studies demonstrate that acupuncture with Chinese herbal medicine relieves clinical symptoms and improves treatment response rates [165]. This combination regulates serum inflammatory factors and reduces gastric mucosal inflammation. For instance, acupuncture with Danggui Sini Tang decreases serum inflammatory markers and suppresses H. pylori-related immune responses, thereby mitigating mucosal damage [166]. The Jianpi Activating Blood Formula, when combined with acupuncture, reduces endoscopic pathological scores while increasing PGI and the PGR. This mechanism may involve regulating pepsinogen secretion, improving glandular function, and inhibiting tissue atrophy [167]. Endoscopic observations reveal that acupuncture with Western medicine promotes the resolution of mucosal hyperemia and edema. This suggests that improvements in microcirculation and tissue nutrition facilitate mucosal repair [168]. Other combinations show promising results: moxibustion with herbs outperforms herbal monotherapy in improving TCM syndromes and gastric mucosal atrophy [169,170]. Acupoint plaster therapy improves mucosal morphology and histological scores, thereby reducing atrophy and the severity of intestinal metaplasia [171,172]. However, most combination studies are single-center trials with small samples, lacking blinding and multicenter validation. Thus, therapeutic efficacy and mechanisms require confirmation through higher-quality research [173].

Additional evidence suggests that acupoint thread implantation combined with quadruple therapy enhances H. pylori eradication rates and symptom improvement. This offers alternatives for drug-resistant infections [174,175]. Acupoint plaster, combined with Chinese medicine, reduces inflammatory mediators, including C-reactive protein, interleukin-6, and tumor necrosis factor-α, thereby suppressing inflammation and promoting mucosal recovery [176]. Needle-knife therapy with thread implantation regulates gastric electrical rhythm and power, potentially through restoring gastric positioning, relieving nerve compression, and modulating vascular-endocrine functions [177]. Auricular therapy with acupoint massage may alleviate gastric pain and anxiety, though evidence remains limited [178].

## 5. Discussion

This review examines the mechanisms of acupuncture and clinical studies for CG, including CNAG and CAG. TCM external therapies—encompassing acupuncture, EA, moxibustion, acupoint catgut-embedding, and acupoint patches—treat CG through multiple mechanisms. These include restoring gastric mucosal function, regulating metabolism and gut microbiota, balancing gastrointestinal hormones, maintaining the equilibrium between apoptosis and proliferation, and providing anti-inflammatory and antioxidant effects. Research reveals distinct intervention focuses: acupuncture primarily modulates gastric mucosal secretion and electrical rhythms, while moxibustion predominantly regulates gastric mucosal blood flow and motility.

This review summarizes the mechanisms and clinical studies of acupuncture for CG, encompassing both CNAG and CAG. TCM external therapies demonstrate five main therapeutic mechanisms: restoring gastric mucosal function, regulating metabolism and gut microbiota, balancing gastrointestinal hormones, maintaining the equilibrium between apoptosis and proliferation, and providing anti-inflammatory/antioxidant effects. Research shows distinct intervention specializations: laser acupuncture primarily promotes mucosal repair and regeneration, while needle acupuncture excels in regulating membrane and neurotransmitter metabolism. Moxibustion demonstrates superiority in regulating energy metabolism compared to acupuncture [63,68]. The most frequently used acupoints across studies are ST36, CV12, ST21, and BL20, with ST36 and CV12 being the most common combination. ST36 regulates inflammatory and oxidative stress factors, restores gastrointestinal hormone balance, and modulates gut microbiota composition [179,180,181]. CNAG may progress to CAG, which increases gastric cancer risk 2–4 times [182,183]. Clinical evidence demonstrates that acupuncture, moxibustion, acupoint patching, and catgut-embedding significantly improve core symptoms, including gastric distension, pain, and acid reflux. These therapies also reduce TCM syndrome scores and modulate molecular mechanisms through multiple pathways: regulating gastrointestinal hormones (GAS, MOT, SS), inhibiting inflammatory pathways (NF-κB, COX-2), reducing pro-inflammatory cytokines (IL-6, TNF-α, IL-1β), and modulating apoptosis/proliferation through PTEN-AKT and JAK-STAT signaling. Additional mechanisms include modulation of gut microbiota, regulation of energy metabolism, and epigenetic influences such as DNA methylation [133].

Although existing evidence supports the potential of external therapies in alleviating symptoms and promoting mucosal recovery in CG, current clinical research faces several limitations: (1) All clinical samples originate exclusively from China. (2) Disease classifications lack standardization. (3) Most trials are small single-center studies. Most clinical evidence fails to meet the standards of evidence-based medicine due to methodological inconsistencies and incomplete follow-up data. Outcome measures, including TCM syndrome scales, demonstrate limited reliability. These design deficiencies restrict clinical standardization of acupuncture for CG. Animal studies primarily focus on CAG models, examining the microscopic regulatory mechanisms involved. Research on CNAG remains limited. Furthermore, environmental factors complicate the translation of metabolic and gut microbiota findings from animal models to clinical applications.

## 6. Outlook and Future Direction

CG, as a multistage progressive inflammatory disease, adversely affects patients’ health and imposes socioeconomic burdens. External therapies like acupuncture offer distinct advantages: (1) They significantly alleviate CG symptoms with high safety, as minor adverse reactions (e.g., transient skin allergies from acupoint patches or mild burns from moxibustion) typically resolve spontaneously. (2) Acupuncture exerts multi-target therapeutic effects through diverse pathways, addressing mucosal damage, hormonal imbalances, and inflammatory responses via distinct mechanisms. (3) Experimental and clinical studies identify ST36 and CV12 as core acupoints for CG. Combining additional acupoints based on symptom patterns and etiology advances precision medicine approaches.

Substantial research exists on TCM diagnosis and treatment of CG. Future investigations should address these key issues to clarify acupuncture’s regulatory mechanisms: (1) While clinical studies on acupuncture for CNAG and CAG are extensive, and animal experiments on CAG have advanced significantly, experimental research on CNAG interventions remains limited. This gap relates to insufficient disease awareness among researchers. Focusing on CNAG progression and acupuncture’s role in preventing mucosal atrophy represents a crucial research direction. (2) Establishing mature, reproducible animal models with optimized designs (e.g., sham-acupuncture controls) is essential. Molecular mechanism studies should integrate multi-omics technologies (single-cell transcriptomics, spatial metabolomics) to explore acupuncture’s complex regulatory networks. Research should identify target cells, molecules, and metabolic pathways through qualitative, quantitative, and spatial approaches. (3) Well-designed, large-sample randomized controlled trials combined with systems biology are needed to elucidate acupuncture’s action networks. Such studies will promote standardized application and a mechanistic understanding of acupuncture in the integrated therapy of CG.

## Figures and Tables

**Figure 1 diseases-13-00363-f001:**
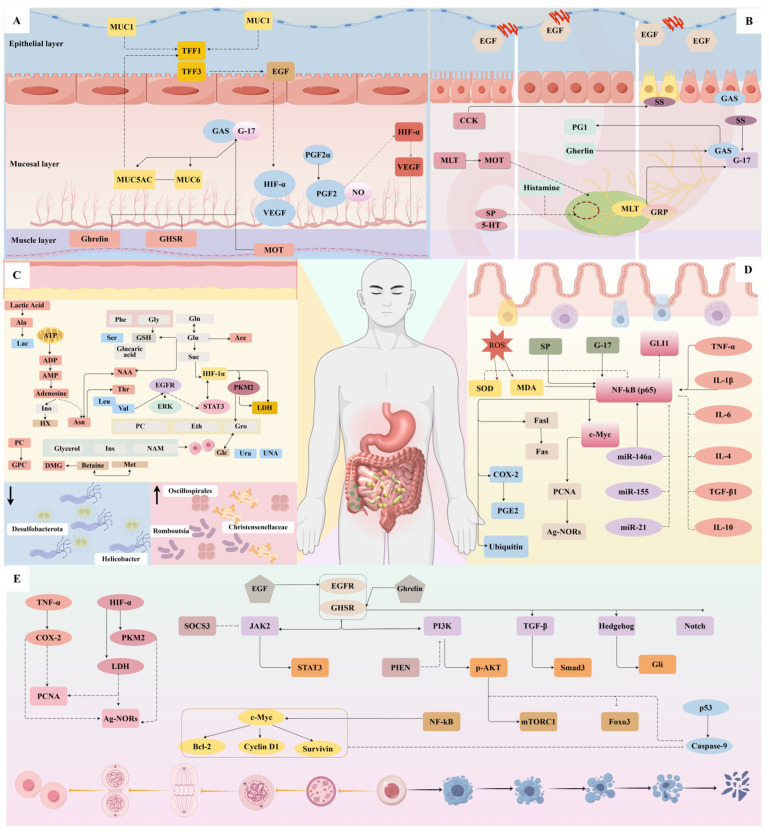
Acupuncture exerts therapeutic effects on CG through five primary mechanisms: (**A**) Regulation of gastric and gastric mucosal functions, where Trefoil Factor 1 (TFF1) and Trefoil Factor 3 (TFF3) facilitate mucosal repair, while Hypoxia-Inducible Factor-1α (HIF-1α) and Vascular Endothelial Growth Factor (VEGF) enhance gastric mucosal blood flow; (**B**) Regulation of gastrointestinal hormones, including Gastrin (GAS), Cholecystokinin (CCK), Somatostatin (SS), Motilin (MLT), Ghrelin (GHR) and Substance P (SP); (**C**) Regulation of metabolism and gut microbiota by inhibiting harmful bacteria and promoting probiotic abundance (↓: Suppression; ↑: Promotion), involving molecules associated with energy metabolism, choline metabolism, and nucleotide metabolism; (**D**) Anti-inflammatory and antioxidant effects through inhibition of pro-inflammatory factors (Interleukin-6 [IL-6], Interleukin-1β [IL-1β], Tumor Necrosis Factor-α [TNF-α]) and the Nuclear Factor Kappa B (NF-κB) pathway, while modulating Superoxide Dismutase (SOD) and Malondialdehyde (MDA) to reduce oxidative stress damage; (**E**) Regulation of cell apoptosis and proliferation through bidirectional mechanisms that inhibit excessive apoptosis for mucosal repair while promoting cancer cell apoptosis to prevent malignant transformation. Key metabolites involved include: Alanine (Ala), Asparagine (Asn), Acetate (Ace), Adenosine Monophosphate (AMP), Adenosine Diphosphate (ADP), Adenosine Triphosphate (ATP), Inositol (Ins), Inosine (Ino), N,N-Dimethylglycine (DMG), Glycerophosphocholine (GPC), Glutamate (Glu), Glycine (Gly), Glycerol (Gro), Glutamine (Gln), Glutathione (GSH), Hypoxanthine (HX), Leucine (Leu), Methionine (Met), Nicotinamide (NAM), N-Acetylaspartate (NAA), Phosphocholine (PC), Phenylalanine (Phe), Succinate (Suc), Serine (Ser), Threonine (Thr), and Valine (Val) (Figure 1). By Figdraw.

**Figure 2 diseases-13-00363-f002:**
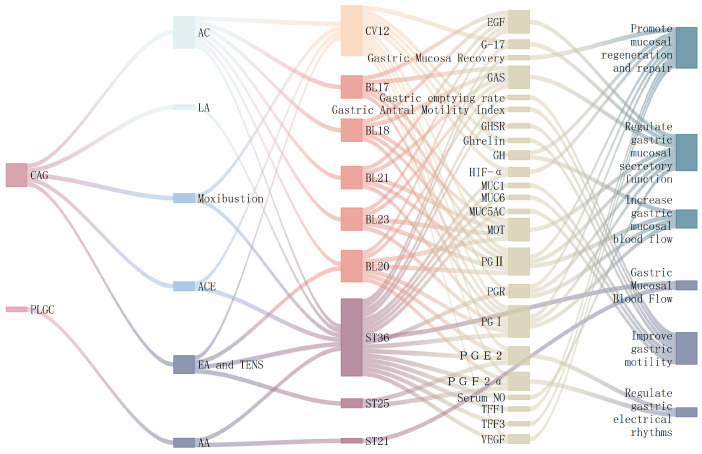
Acupuncture Treatment for CG: Efficacy, Mechanism of Action, and Gastric Function-Related Network. In the Sankey diagram, the width of each node represents its frequency of mention within the entire process network. Both the link width and node size directly correspond to the number of mentions for each pathway, as presented in Table 1, using OriginPro 2024b software. The inclusion of intervention methods: Acupuncture (AC), Laser Acupuncture (LA), Acupoint catgut embedding (ACE), Transcutaneous Electrical Nerve Stimulation (TENS), Acupoint Application (AA).

**Figure 3 diseases-13-00363-f003:**
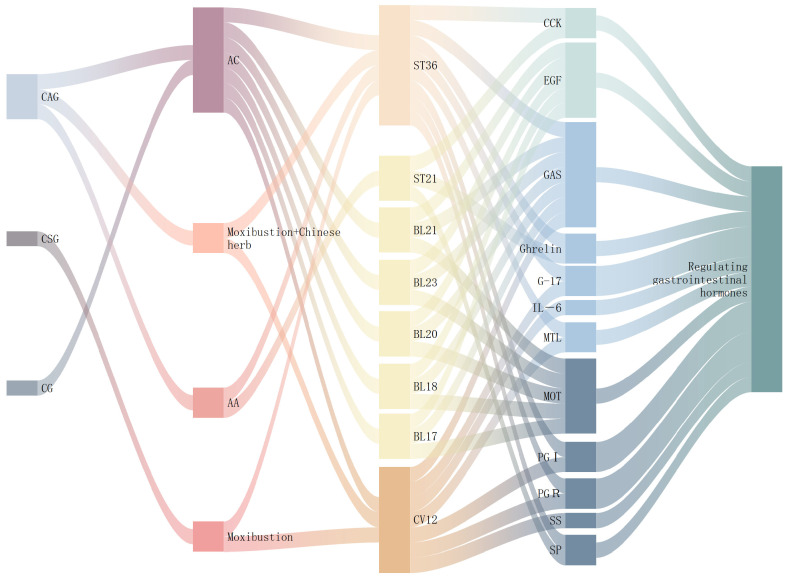
The association network of gastrointestinal hormones in the study of the efficacy and mechanism of acupuncture treatment for CG. In the Sankey diagram, the width of each node represents its frequency of mention within the entire process network. Both the link width and node size directly correspond to the number of mentions for each pathway, as presented in Table 2 using OriginPro 2024b software.

## Data Availability

Not applicable.

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
