# Peer review of "Research Progress on the Efficacy and Mechanism of Acupuncture in Treating Chronic Gastritis"

_diseases, 2025, doi:10.3390/diseases13110363_

Round 1

Reviewer 1 Report

Comments and Suggestions for Authors

The report is attached

Comments on the Quality of English Language

The manuscript is written in adequate but sub-standard academic English for publication. The overall comprehension is possible, yet the writing requires thorough professional editing to ensure clarity, concision, and correct scientific tone. The prose frequently uses long, compound sentences that mix several ideas without clear hierarchy, which makes paragraphs heavy and occasionally ambiguous. For example, mechanistic discussions often combine physiological background, animal results, and interpretive commentary within a single sentence. Breaking these into shorter, logically ordered statements would greatly improve readability.
Verb tense is inconsistent throughout. Methods and literature-search descriptions alternate between present and past tense (“this study collects…,” “we searched…,” “the experiment showed…”). For a review, it is preferable to use past tense when describing what individual studies found and present tense when discussing established concepts (“Acupuncture has been shown to influence…”).
Articles and prepositions are often missing or misused-for instance:
-“in presence of inflammation”  should be “in the presence of inflammation”;
-“by regulating gastrointestinal hormone”  should be “by regulating gastrointestinal hormones.”
Such grammatical inaccuracies are minor individually but occur frequently enough to interrupt the professional flow expected in an MDPI paper. The tone occasionally overstates the evidence base, using assertive verbs such as “demonstrates,” “proves,” or “confirms” where the cited data are preclinical or small-scale. These should be replaced by more cautious expressions: “suggests,” “indicates,” “is associated with,” or “has been observed to.” Maintaining an objective, evidence-weighted tone is essential, especially in reviews dealing with traditional or complementary therapies.
Punctuation and formatting require correction: commas are inconsistently used; spaces are missing before parentheses (e.g., “Electroacupuncture(EA)” should read “Electroacupuncture (EA)”); and there are inconsistencies in abbreviation formatting (“GAS, MTL, SS” sometimes appear with varying capitalizations or extra spaces). Terminology should be standardized across the text; for example, use one consistent form for each modality (e.g., “electroacupuncture (EA)” throughout, not alternating with “EA therapy”), and for disease abbreviations (“CG,” “CAG”).
The Abstract and Conclusions contain phrases that over-interpret results, such as claims of “preventing progression to gastric cancer.” These statements exceed the evidence and should be rewritten in conditional or exploratory language to match the data cited. Figure legends and captions are generally understandable but could be improved with clearer grammatical structure and consistent capitalization. Overall, the English proficiency level can be classified as upper-intermediate; sufficient for conveying scientific ideas but not at publication quality.

Author Response

Dear Reviewer 1,We sincerely thank you for reviewing our manuscript. We greatly appreciate your constructive feedback on this study and have incorporated your suggestions into the revised version. All modifications in this revised manuscript are highlighted in red text. Our point-by-point responses to each of your comments are provided below. Please refer to the uploaded file for details.

Reviewer 2 Report

Comments and Suggestions for Authors

This narrative review is rich in descriptive content and illustrates acupuncture’s multi-target potential in chronic gastritis. However, it has multiple concerns that must be addressed:

  1. The article compiles studies without assessing their methodological rigor, sample size adequacy, or experimental controls. Consequently, strong and weak studies are treated with equal weight, limiting interpretative validity.
  2. The majority of cited evidence is derived from rat or mouse models of chronic gastritis, with minimal reference to randomized controlled trials in humans (restricts clinical generalizability)
  3. Despite citing multiple studies, the paper offers no pooled data, statistical analysis, or effect-size comparison to support claims of efficacy.
  4. Most cited studies appear to originate from Chinese databases (e.g., CNKI, Wanfang, VIP), increasing the risk of language bias and exclusion of relevant international research.
  5. The search process lacks specifics on time frame, inclusion/exclusion criteria, and search terms for each database, reducing transparency and reproducibility.
  6. The paper attributes biological improvements solely to acupuncture, with little consideration of placebo effects, concurrent medications, or spontaneous healing in chronic gastritis models.
  7. While mechanisms are well summarized, there is little information about how the included studies were designed (randomization, blinding, controls), which is crucial for evaluating internal validity.
  8. The authors often extend results from narrow preclinical models to broad clinical conclusions (e.g., “acupuncture can prevent cancer progression”), which overstates current evidence.
  9. The review selectively highlights studies showing positive effects of acupuncture but omits research reporting neutral or negative outcomes.
  10. Several sections (e.g., on hormone regulation and oxidative stress) repeat similar mechanisms with minimal synthesis, plus repetition
  11. Each mechanistic pathway (metabolism, apoptosis, inflammation) is discussed in isolation; the review lacks an overarching model that interlinks these domains into a cohesive conceptual framework.
  12. The paper focuses heavily on cellular and molecular mechanisms without adequately linking these effects to patient-centered outcomes such as symptom relief, recurrence rates, or quality of life.
  13. The review groups diverse techniques (electroacupuncture, moxibustion, laser, catgut embedding) together without evaluating their relative efficacy or consistency across protocols.
  14. No discussion is provided on safety profiles, side effects, or procedural risks of acupuncture interventions—an essential component of clinical relevance.
  15. The conclusion reiterates known benefits without offering a clear summary of research gaps, methodological recommendations, or translational pathways toward clinical practice.

Author Response

Dear Reviewer 2,We sincerely appreciate your valuable and constructive feedback. We have carefully revised the manuscript and provided a point-by-point response below. We hope that your comments have been addressed accurately. The revised sections are highlighted in red, and our responses are presented in red text. Please refer to the uploaded file for further details.

Reviewer 3 Report

Comments and Suggestions for Authors

This review examines the therapeutic and mechanistic roles of various acupuncture modalities, including electroacupuncture, moxibustion, acupoint catgut embedding, and acupoint application, specifically in the management of chronic gastritis (CG) and chronic atrophic gastritis (CAG).  Although the topic is pertinent and holds potential significance for the journal's readership, Diseases, the current manuscript version exhibits notable methodological and interpretative limitations. The article predominantly presents as a narrative summary rather than a rigorously structured scientific review. There is insufficient transparency regarding the literature search methodology, inclusion criteria, and the framework for evidence appraisal. While the synthesis of mechanistic insights is thorough, it frequently veers into descriptive repetition, lacking a critical assessment of study quality and its clinical implications.

Major Comments

The review lacks specific details regarding the literature search strategy, including the databases searched, keywords used, time frame, inclusion/exclusion criteria, and any language filters applied. Then, what databases and search terms were used to identify the included studies?

It is essential to clarify whether this review is intended to be narrative, systematic, or scoping in nature, and to adhere to the appropriate reporting guidelines (e.g., SANRA or PRISMA-ScR).

How many clinical trials versus animal experiments were ultimately reviewed? The majority of cited studies are preclinical or animal-based, with limited clinical evidence presented and not adequately analyzed.

Are there standardized acupuncture protocols (frequency, duration, acupoints) used across studies? It is essential to distinctly separate clinical studies (such as randomized controlled trials and observational studies) from preclinical research, highlighting their endpoints, effect sizes, and inherent limitations.

Additionally, is there evidence of dose–response or modality-specific differences between electroacupuncture and moxibustion?

The review amalgamates different types of interventions under the umbrella term "acupuncture," while the biological mechanisms can vary significantly among modalities such as electroacupuncture, moxibustion, and laser or catgut embedding.

Consider reorganizing the content into separate subsections for each modality, detailing parameters (e.g., frequency, duration, acupoints, and dosage of heat/light stimulation) alongside the strength of corresponding evidence.

Statements like “acupuncture offers significant advantages in treating chronic gastritis” are not substantiated by comparative data against standard therapies (such as proton pump inhibitors or H. pylori eradication).

The conclusion should stress that while mechanistic insights are encouraging, they remain predominantly preclinical and necessitate validation through high-quality randomized controlled trials.

Although the manuscript enumerates several pathways (e.g., NF-κB, PI3K/AKT, TGF-β/Smad, JAK/STAT) and molecular targets, it fails to connect them to functional or clinical outcomes.

Incorporating a summary table or figure that relates these signaling cascades to measurable effects (such as histology, mucin expression, inflammatory markers, and symptom relief) would enhance readability and the translational impact of the findings.

The review summarizes results without critically evaluating sample sizes, study designs, reproducibility, or species differences.

Furthermore, there is an absence of discussion regarding safety and adverse events, which is crucial for clinical application.

What are the main gaps that future multicenter RCTs should address?

Minor Comments

Use consistent terminology (CG vs. CAG; EA, MA, LA).

Define all abbreviations at first mention (e.g., PGI, PGII, GAS, MTL, G-17).

Provide numeric or percentage data where available (changes in biomarker levels, histological scores).

Standardize table formatting and cite all references directly in figure/table captions.

Some sentences are repetitive or vague—condense where possible for clarity.

Author Response

Dear Reviewer 3, thank you for reviewing our manuscript and providing constructive feedback. Your valuable suggestions have significantly contributed to enhancing the quality of our work. We have meticulously revised the manuscript and addressed each comment point by point below. We hope that our responses accurately reflect your concerns. Revised sections are highlighted in red, and our responses are presented in red text. Please refer to the uploaded file for further details.

Round 2

Reviewer 1 Report

Comments and Suggestions for Authors

The second report is attached.

Comments on the Quality of English Language

The English of the revised manuscript is improved compared with the previous version, but it is not yet publication-ready. The text still reads like a direct translation from Chinese academic prose: sentences are long and layered, the subject is sometimes far from the verb, and connectors are overused. Several passages reproduce the same syntactic frame 3-4 times in a row (“…thereby…,” “…thus…,” “…therefore…”), which makes the text heavy. There are also recurrent micro-issues:
-article use (“the chronic gastritis” when it should be “chronic gastritis”),
-prepositions (“in the regulation of,” “on the effect of,” “in both excitatory and inhibitory ways” could be simplified), and
-number/distribution agreement (“the regulated gastrointestinal hormones being relatively consistent…” should be “the gastrointestinal hormones regulated by acupuncture were broadly consistent across studies”).
The style can be made suitable for the journal by:
-splitting long methodological/mechanism sentences into 2-3 shorter sentences;
-standardizing tense: present for general statements, past for individual studies;
-replacing literal phrasing with conventional review language;
-running a final professional edit.

If these steps are followed, I think the English will be fully adequate for publication.

Author Response

Dear Reviewer 1: We sincerely thank you for reviewing our manuscript. We greatly appreciate your constructive feedback on this study and have incorporated your suggestions into the revised version. All modifications in this revised manuscript are highlighted in red text. Our detailed responses to each of your comments are provided in the document.

Reviewer 2 Report

Comments and Suggestions for Authors

All corrections are satisfactorily made

Author Response

We would like to sincerely thank you for your time to read and to assess our manuscript and to provide us with your comments and suggestions to improve it.

Reviewer 3 Report

Comments and Suggestions for Authors

The authors provided a detailed response letter. Consistent changes were performed in the manuscript. I am favorable with its acceptance in the current form.

Author Response

(The authors gave the same response as above.)
